# Women's strategies for navigating a healthy sex life post-sexual trauma

**Kristen P. Mark** [1,2]*, **Laura M. Vowels** [3], **Lindsey Mullis** [4], **Katarina Hoskins** [2]

**1** Department of Family Medicine & Community Health, University of Minnesota Medical School, Minneapolis, MN, United States of America, **2** Institute for Sexual & Gender Health, University of Minnesota Medical School, Minneapolis, MN, United States of America, **3** Department of Social and Political Sciences, Family DevelOpment Lab (FADO), Institute of Psychology, University of Lausanne, Lausanne, Switzerland, **4** Human Development Institute, University of Kentucky, Lexington, KY, United States of America

* kpmark@umn.edu

## Abstract

Sexual trauma is common. Consequences of sexual trauma can include deterioration of mental and physical health and it can also affect future romantic and sexual relationships. Previous studies have identified common healthy and destructive coping mechanisms to recover after experiencing sexual trauma, but few studies have investigated useful strategies to move into a healthy sexual relationship focused on resilience. In-depth semi-structured interviews were conducted with 41 women with a history of sexual trauma who were in a healthy sexual relationship at the time of participation. Participants provided strategies that helped them move beyond the sexually traumatic event(s) toward a healthy sexual relationship. Reflexive thematic analysis identified 5 effective and 6 ineffective strategies reported by the participants. Rich examples of resilience and empowerment were overarching in the effective strategies used for moving toward healthy sexual relationships. Women were also able to reflect on the strategies that were ineffective for them with kindness and understanding for their coping at that time, a normalizing theme for women working through sexual trauma. The results of this study will help therapists and researchers working with women who have experienced sexual trauma learn from their experiences in working beyond trauma toward a healthy sexual relationship.

## Introduction

In recent years there has been a significant shift toward more open conversation related to sexual assault and sexual trauma. Momentum from campaigns such as #MeToo and #TimesUp are giving more attention to the gravity of sexual assault and trauma. As an increasing number of women speak out about their experiences, a clear need has emerged to address the frequency and potential negative consequences of sexual trauma in addition to the process of healing post-sexual trauma. Definitions of sexual trauma vary; however, the current study defines sexual trauma as physical or emotional trauma resulting from a sexual act committed against someone without their consent. Data collected by the National Intimate Partner and Sexual Violence Survey [1] indicate that one in five women in the United States will experience sexual assault in her lifetime. Sexual trauma that results from sexual assault can seriously impact a

**Data Availability Statement:** Since these are sensitive data with potentially identifying information and since the participants only consented to participate based on the knowledge that we would be the sole viewers of their data and

their data would not be disseminated individually but only in the context of summarization with others' data, we are not able to provide these data as publicly available. Since these are qualitative interview data, there is not a "minimal data set" available, as there are no numerical data associated with the dataset. For a non-author contact for any future concerns regarding data, please reach out to the primary ethics committee that the study was approved through, University of Kentucky Office of Research Integrity via phone (859-257-9428) or email (rs_ORI@uky.edu).

**Funding:** The authors received no specific funding for this work.

**Competing interests:** The authors have declared that no competing interests exist.

person's mental, physical, and emotional well-being and can affect future romantic and sexual dynamics [2–4]. Although several studies look closely at the impact sexual trauma can have on an individual [3–6], these studies do not provide an in-depth examination of specific strategies used by women to not only cope with their trauma but move beyond it toward healthy and happy sexual relationships. In the present study, we interviewed 41 women who had experienced sexual trauma and reported currently being in a healthy relationship to better understand the strategies used to recover from the trauma and move on to a healthy sexual relationship.

Many survivors of sexual assault experience symptoms of trauma such as hypervigilance, avoidance, and re-experiencing because of the assault [5–7]. These sexual trauma symptoms are often associated with negative outcomes such as post-traumatic stress disorder (PTSD), anxiety, depression, substance use, aggression, and self-injury [5–12]. A recent meta-analysis found that sexual assault was strongly associated with PTSD [13] and a qualitative review found that between 17% and 65% of sexual assault survivors develop PTSD [14]. Survivors of sexual assault often experience negative social reactions following the assault, further contributing to symptoms of PTSD [15]. Symptoms of post-traumatic stress can have serious consequences on impairment across multiple domains of functioning such as additional physical health problems, revictimization, problems with alcohol or drug consumption, and an increase in negative social reactions [7, 15–18]. Moreover, individuals who have experienced previous stressful life events, such as childhood sexual abuse (CSA), are at greater risk of developing trauma symptoms from subsequent traumatic events [19, 20].

In addition to mental health problems, especially PTSD, sexual assault is often associated with a decrease in sexual functioning and satisfaction [12, 21–26]. Early studies on sexual assault survivors found that most of the women who had a history of sexual trauma (71% and 88.2%) attributed their sexual problems, including fear of sex, desire difficulties, or arousal dysfunction, to their assault [21, 22]. A more recent survey of college women who survived sexual assault reported similar levels of sexual dysfunction (87%) including a decrease in sexual desire and difficulty achieving an orgasm; they were also more likely to experience multiple sexual dysfunctions [24]. Some of the negative effects of sexual assault on sexual functioning including fear, arousal, and difficulties with desire can persist for years [12]. A recent qualitative study of 45 sexual trauma survivors and their primary informal support providers (e.g., partner, friend, family member) found both an increase and a decrease in interest and participation in sexual behavior. When the sexual behavior increased in frequency, it often involved engagement in sex work or cycling through many sexual partners [25]. Together these studies highlight the extent of the impact sexual trauma can have on the survivor's future sexual and relational functioning and the importance of better understanding how to mitigate the deleterious effects of sexual trauma on survivors.

The impact of exposure to sexual trauma can be dependent on the presence of PTSD symptoms [27]. Individuals who developed symptoms of PTSD following sexual assault experienced significant decline in some aspects of sexual functioning including sexual aversion, sexual pain, and sexual satisfaction but did not differ on frequency of orgasm or sexual arousal. In contrast, participants who had experienced sexual trauma but did not develop symptoms consistent with PTSD did not significantly differ in their level of sexual functioning from participants who had never experienced sexual trauma [27]. These conclusions highlight the importance of better understanding the strategies and resources that some trauma survivors may have that enable them to recover from the trauma to move toward healthy relationships.

As outlined above, sexual functioning and sexual satisfaction are often impaired following sexual trauma. We know from previous literature that sexual and relationship satisfaction are bidirectionally strongly connected [28, 29], thus it is likely that individuals who experience a

decline in sexual functioning and sexual satisfaction are also likely to experience a decline in relationship satisfaction. Indeed, previous studies have found that sexual trauma can impact relationship functioning and dissolution [30, 31]. Fear of re-traumatization often leads to isolation and avoidance, further affecting social dynamics for the individual [6]. Following a disclosure, many survivors and their partners report a change in the relationship, both with positive and negative effects. Partners of survivors sometimes felt their needs were not being met and they experienced strain in the relationship following disclosure of sexual assault [25]. However, although disclosure of prior sexual trauma to a new partner is common, it isn't always supportively received, and the reception of the disclosure may impact the extent to which the relationship experiences strain [32]. In another study, the results showed that partners were cautious about navigating sexual intimacy with the survivor and survivors reported that they sometimes got triggered by their partner or experienced dissolution of their relationship because of the impact of the assault on sexual functioning [25]. These results suggest that sexual trauma is likely to have a wider impact on the survivors' relationships in addition to individual functioning.

The impact sexual trauma has on overall health and well-being indicates a crucial need for more information addressing how to cope with sexual trauma and especially how to navigate future sexual relationships post-sexual trauma. While a great deal of research has addressed the potential impact of sexual trauma on individuals' functioning, surprisingly little research has been conducted on how to minimize the negative impact of sexual assault on the survivors' well-being. Research addressing the implementation of therapy-related coping techniques on those with trauma indicates that inconsistencies and incompatibility with these techniques can retraumatize the individual [33]. Although some research examines coping strategies implemented or encouraged for those who have experienced trauma, most of the existing research has focused primarily on therapy and cognitive techniques [33–35] these have been shown to be valuable mechanisms. Yet little is known about additional coping strategies used by individuals to move forward to healthy relationships and cope with ongoing symptoms of sexual trauma throughout their lifetime.

## The current study

Chanel Miller, sexual trauma survivor and author of the book 'Know My Name: A Memoir Chanel Miller' [36] emphasizes the importance of society to address sexual trauma and hold space for survivors. Miller also expresses the value of taking on simple strategies to cope with trauma in daily life. Previous research has addressed the gravity of the impact left by sexual trauma [5–7, 11], but much of the research into treatment of sexual trauma has focused on developing treatment protocols for therapy. However, the extant literature lacks a more comprehensive understanding of the range of strategies survivors can use to mitigate and combat their trauma.

As women are disproportionately affected by sexual trauma [5], women may have more extensive strategies they use to cope with trauma and are particularly poised to give insight. The current study aims to fill a clear gap in the literature by addressing the question of how women move past sexual trauma and into a healthy relationship. We do this by giving voice to a sample of women across a large age range (18 to 55) to identify specific strategies that they used, effective and ineffective, to move beyond their experience of sexual trauma and into healthy relationships.

## Method

### Participants

In this semi-structured interview study, we recruited 41 women to participate in a phone interview that lasted an average of 42 minutes (range of interviews was 28 minutes to 78 minutes).

Data collection concluded once we reached data saturation and repetition began to emerge from the interviewees. Participants were recruited primarily through online advertisements placed on Twitter and Facebook relying on network snowball sampling, with some advertisements posted in cafes and around a mid-sized university campus in a medium sized city in the United States. Most participants resided in the United States ($n = 35$; 85.4%), with a minority residing in Canada ($n = 3$; 7.4%), Australia ($n = 1$; 2.4%), England ($n = 1$, 2.4%), and New Zealand ($n = 1$, 2.4%). Participants were eligible to participate in the study if they identified as a woman, were over the age of 18, had a history of sexual trauma (self-defined), and were currently in what they considered a healthy sexual relationship. In the recruitment process, we screened for sexual violence in their current relationship by asking participants if the sexual trauma they experienced happened within their current relationship and if they have ever felt physically or emotionally unsafe in their current relationship. None of the participants indicated sexual violence in their current relationship.

Participant ages ranged from 18 to 55, with an average age of 29. A little more than half of the sample identified as heterosexual ($n = 27$; 65.8%), with a minority of participants identifying as bisexual ($n = 5$, 12.2%), lesbian ($n = 1$; 2.4%), pansexual ($n = 2$; 4.9%), queer ($n = 4$; 9.8%), and questioning ($n = 2$; 4.9%). All but two of the participants were in a relationship with a man at the time of data collection (one participant was in a relationship with another woman and the other was in a relationship with a non-binary partner) and the average relationship length was 50.78 months (4.23 years) with a range from 3 months to 29 years. A minority of our participants ($n = 3$, 7.3%) indicated that they were in a polyamorous relationship. Many of the participants were self-described as White ($n = 33$, 80.5%), though 8 (19.5%) self-identified as a racial or ethnic minority (self-described as one of the following: AfroLatina, Asian, Black, Filipino, Jewish, Hispanic). Most of the participants experienced sexual trauma after the age of 12 (71%), with 25% having experienced sexual trauma before the age of 12 and 4% who reported sexual trauma having happened both before and after the age of 12.

## Measures

The current paper was part of a larger study aimed at understanding how prior sexual trauma experience informs navigating communication, consent, sexual pleasure, and other aspects of romantic relationships in women. This was done through the iterative analysis of answers to the following questions: "What does sexual consent look like in your current relationship?", "Do you feel safe voicing your sexual wants and needs?", and "What strategies have been helpful or unhelpful in navigating this?" The answers provided to these questions were analyzed specifically for the effective and ineffective strategies implemented to navigate consent and sexual agency in their current relationships.

## Procedure

When participants were exposed to the advertisement for the study that clearly asked for women who had experienced sexual trauma but were now in a healthy sexual relationship(s), they were instructed to email the first author to schedule an interview where they were screened for eligibility. If the participant met eligibility criteria (provided above), they were consented into the study with a verbal informed consent and they were informed the interview would be recorded for later verbatim transcription with all identifying information masked. All interviews were conducted by the first author over the phone regardless of geographical location to provide methodological consistency. Participants were free to skip any question that made them uncomfortable or that they did not want to answer. At the end of the interview, participants were provided with a $20 online gift card as appreciation for their

participation and they were provided the opportunity to access a list of resources in case the interview brought up any feelings for them that needed further processing. All interviews were audio recorded and transcribed verbatim. Any identifying information from participants were removed from the transcripts and names were replaced with pseudonyms. All protocol were approved by University of Kentucky's Institutional Review Board (IRB #46482).

## Analytical approach

In this study, we followed reflexive thematic analysis taking an inductive approach based on Clarke and Braun [37] to reflect the explicit content of the data provided by the women. Authors familiarized themselves with the data before creating a series of codes that were identified from the interviews to provide insight to the research questions of interest. These codes were then classified into themes and an iterative process was followed between coders in generating, reviewing, defining, and naming themes from the codes generated by the data. Any disagreements between classification of codes were discussed until 100% agreement could be reached. Although the transcripts were taken and analyzed as a whole, the classification of codes into themes was conducted separately for our research questions. The analytic process resulted in the identification of three themes in the strategies used to establish consent in their current relationship and six themes in the evolution of the sexual agency to communicate needs and wants post sexual trauma. We also had an auditor familiar with the study and thematic analytic technique but not familiar with this particular research question read through the results and assess any inconsistencies with the themes or thematic representative quotes. We utilized '[. . .]' signals to remove unnecessary detail or provide needed additional information in the quoted data provided. All identifying information was masked in quotes provided to support the themes all names used in the labeling of the quotes are pseudonyms. Additionally, spelling, grammatical, and typographical errors were all corrected to aid readability in the quoted data.

## Results

Five themes were identified in the data that demonstrated effective strategies expressed as coping mechanisms that promoted participants' progress toward moving beyond sexual trauma and into healthy relationships. Additionally, six themes were identified as ineffective strategies for moving beyond sexual trauma into healthy relationships. We outline each of these below and include an overarching quote that best exemplifies each identified theme. Additional quotes from the data illustrating the themes are outlined in the S1 Table.

### Effective strategies

The five effective strategies identified in the data were strategies that participants perceived to be helpful in moving them beyond sexual trauma into healthy and happy sexual relationships. They included: social support, communication, self-care, therapy, and empowerment; each are detailed below.

**Social support.** The most identified effective strategy theme was social support as a valuable aspect of healing. Importance was placed on finding trusted sources to communicate with and share one's story. This was most often exemplified through finding a community of individuals with shared experiences, and this was both empowering and comforting for the participants in our study. Seeking out shared experiences included volunteering or working at rape crisis centers, women empowerment organizations, and surrounding oneself with an understanding crowd while having the opportunity to offer support to others. An example of social

support through the sense of community was provided by Jess, a 26-year-old AfroLatina participant:

"So, one [effective strategy] is the circles that I surround myself with. . . giving me the freedom or space to grow and to think about the things that I've experienced in my own life through hearing other people's situations. An example of what I'm talking about specifically is I'm involved in a women's organization. We meet and support each other through either current or past situations that we have encountered in our lives and being part of that group has been very healing for me."

Another demonstration of social support was reflected in the participants' emphasis on the importance of talking to the right people, recognizing the impact of social support, and relying on individuals that provided a positive response in their supportive role. This included the need to "weed out" those deemed as unworthy of hearing one's story and working to "get your tribe" and finding the social supports that provided positive encouragement. The following quote from Mel, a 28-year-old white participant, illustrates this nicely:

"I feel wildly lucky to have a huge network of friends, especially friends that have sexual trauma histories. At any time, there are at least four women that I can step out and call and go on a walk on the phone like this and get that support. Some of the people that I care about aren't connected to big networks like this, and wow, that would be so hard to be dependent on your primary male partner for your emotional support."

**Communication.** Communication was also important to the healing process and moving toward healthy sexual relationships. Communication was important for who and where to share information with, but also in clearly communicating boundaries in relationships or sexual settings. Communicating about their experiences was a healthy and beneficial way for the women to process the events and move forward as exemplified by the following quote from Margo, a 29-year-old white participant: "Actually just talking about it, both with friends and with my partners has been, particularly with my current partner, it's been really helpful in my healing process to be able to talk very frankly and openly." Communicating boundaries involved the development of physical and emotional boundaries and learning to better communicate those boundaries in various settings and relationships. Importance was placed on the women self-identifying those boundaries with additional emphasis on the ability to clearly communicate those boundaries to others, as depicted in this quote from Patti, an 18-year-old white participant:

"Be explicit about your boundaries and figure out. . . It's just trial and error. Sometimes it's random things that you wouldn't even think would be triggers or things like that. As you go through and you just start to learn these are things that I get freaked out about, these are triggers for me, and just working through that and making sure that your partner knows those boundaries."

**Self-care.** Self-care was identified by the participants as a critical component to the healing process and was characterized by time and activities dedicated to oneself. This included physical and psychological outlets including exercise, yoga, relaxation, and mindfulness practices. Participants expressed opportunities where they focused on engaging in activities that promoted a healthy well-being, as Jane, a 33-year-old white participant explains: "I also did a lot of, I guess spiritual therapy. I did a lot of tantra. So, I think that really helped me as well to

reconnect with my body and reconnect with my sense of pleasure." An additional demonstration of self-care involved emotional and creative outlets such as art, writing, journaling, and reading. These self-care practices provide opportunities to focus dedicated time to themselves and their individual needs, thoughts, or physical self with a creative modality or outlet. The following quote from Jen, a 24-year-old white participant demonstrates this as a successful strategy for coping:

> "I have also found therapy in journaling. I think that just is a helpful kind of low-cost way to kind of get that out of the box that's in our head and get them out on paper and so they're not necessarily there suffocating us."

**Therapy.** Several participants discussed the role that formal therapy played in their path toward healthy sexual relationships after experiencing sexual trauma. Feeling open and having access to take this step toward formal healing provided a space for processing that was not easily accessed outside of a professional setting and provided a structured communication for those learning to experience and express feelings about their experiences. Therapy provided an opportunity for growth that was significant, as noted by Cait, a 22-year-old white participant: "The most important thing that I had ever done for myself, was put myself in therapy. . . If you had this conversation with me before I put myself in therapy, it would have been very different."

**Empowerment.** The theme empowerment was defined by self-reflection, self-awakening, self-advocacy, and self-love or appreciation. This connects to previously mentioned themes such as social support, self-care, and therapy as the women found ways of feeling empowered through these strategies. The feeling of empowerment motivated the continuation of practicing successful coping strategies such as gaining education and knowledge about abuse, learning self-defense techniques, focusing on loving oneself and building a positive self-image, along with working to help others through their own experience. Empowerment in this context centered around a sense of taking ownership and holding onto a sense of control. This is best represented in this quote from Katarina, a 22-year-old white participant:

> "As far as self-help. . . let's see. It keeps bouncing back to the self-empowerment, so ways that I make myself feel empowered. I have specific self-care days. That is very important for feeling good, empowered, and like I'm in control of my life. If I don't feel in control of my life, then any of my mental health symptoms will come up, which includes that generally not feeling good about the things that happened during my sexual traumas."

Reframing was a tool used by participants to fuel their self-empowerment. This involved letting go and engaging in forgiveness, which facilitated a process of self-love and reflection through reframing away from victimization and moving toward acceptance. Undergoing this process and working toward self-love and acceptance provided space for healing. Recognizing that one cannot change the past but instead can enact control over how one thinks and feels into the future is an individual approach for empowerment while processing a view of acceptance for the experience. Jos, a 25-year-old white participant provides an example of how reframing was used as a successful strategy of empowerment:

> "The other thing that's been helpful to me is really thinking about it as 'I can't change what happened, but I can decide what it means to me going forward', meaning I can use it to sort of work to protect others."

## Ineffective strategies

We identified six ineffective strategies in the data, and these were defined as strategies that turned out to be negative, unhealthy, and/or unhelpful in moving forward from sexual trauma to a healthy sexual relationship. Kate, a 37-year-old white participant highlighted several of these ineffective strategies in her experience, characterized as:

> "So, the most unhealthy [coping mechanisms] were at that time, being extremely promiscuous, that was. . . yeah, that's actually just been a theme most of my life, but self-harm and cutting, drug abuse, substance abuse, drinking. Anything that I used I overused as a means of escape. Those were all very unhealthy for me."

**Casual sex.** Engaging in casual sex was used as coping mechanism to 'feel something' perhaps with the intent to feel empowered but overwhelmingly the interviewees who mentioned this theme reflected negatively on this strategy and suggested that others not engage in casual sex as a coping strategy. This is exemplified by this quote from 42-year-old white participant Nicole: "Yeah. I mean, for me, I went through a brief phase where I was like, 'I'm going to be sexually empowered and hook up with guys,' and that was horrible. Do not recommend." This doesn't mean that casual sex is inherently bad, but that when using it as a coping mechanism to numb or escape, it was not particularly helpful for the participants in getting to a healthier place or forming a healthier relationship with sex after experiencing sexual trauma.

**Unhealthy communication.** Communicating to unsupportive people, over-communicating with less than trustworthy sources, and a general lack of social support was identified as a common theme in the data for an ineffective strategy. Interviewees referenced the negative reactions of others and feeling as though they were unsupported or were even made to feel as though they were being untruthful about their traumatic experience. In this sense, communication could be painful and did not aide the process of moving forward. An example that illustrates unhealthy communication and lack of social support was provided by Jess, a 26-year-old AfroLatina participant: "Finding trustworthy people to confide in can be really beneficial. I had unfortunately confided in people who I didn't feel like they believed me. And I didn't feel like they supported me."

**Avoidance.** Often due to experiences with unhealthy communication or lack of social support, participants reported that they wasted a great deal of time avoiding healing from sexual trauma, and this ended up being a barrier to moving on to a healthy sexual relationship. Women avoided certain conversations and encounters that left them feeling more and more isolated over time. Disassociation aided in momentary relief but was not a beneficial long-term strategy. Several of the women mentioned dissociating during sexual encounters and becoming highly distracted from reality. Isolation via self-sabotage, avoidance of communication, and simply not addressing the trauma in a productive manner was all encompassed in this unhealthy strategy. This avoidance is exemplified in the following quote from Margo, a 29-year-old white participant:

> "Coping by dissociation was effective in the moment and not helpful long term. And some of that's like for a little while it was an automatic response, and then it was like a practice plan. I was like, 'Alright, time to check right out'. That was effective in the moment and just also didn't contribute to a fulfilling relationship or to a fulfilling experience of sexual pleasure."

**Substance use.** One strategy that aided in avoidance but was identified as a unique theme in the data was the use of substances to numb or escape. This was characterized by the use or

abuse of drugs and alcohol. The women who employed these strategies stated clearly that masking emotions and using alcohol or drugs to escape or cope simply did not help. This again could provide a momentary escape but did not promote long-term solutions in coping with the sexual trauma and progressing toward healthy relationships. An example of substance use as a strategy that did not work is depicted in this quote from Mindy, a 26-year-old white participant:

> "Other things that I wouldn't recommend, I definitely wouldn't recommend trying to cope with drugs or alcohol. I mean, trust me. I understand the pain that comes with it. But the pain that comes with trauma I should say, but drugs and alcohol aren't going to help resolve those issues. They're not gonna bring a person to a better place. They're just going to mask everything, and I'm not knocking that. Sometimes you just need that. Sometimes you just need a time out from your emotions, but that's definitely not a viable longer-term solution."

**Therapy.**    Although therapy was identified as an effective strategy, there were also several participants who cited therapy as an obstacle to healing. This highlights the importance of having a therapist who is a strong fit with the needs of the client. This theme was characterized by women discussing certain treatment approaches that the women felt were not the most appropriate fit for their needs. A primary complaint regarding therapy was the lack of diversity and inclusivity among professionals. Several of the women felt underrepresented and therefore misunderstood when attempting therapy and moved on to use different strategies. This quote from Mel, a 28-year-old white participant, demonstrates therapy as an ineffective strategy:

> "Yeah, so I have seen a couple of therapists here and there without too much luck. I mean, I just feel like being a very loud, clear sex worker is not exactly middle-class therapist friendly. There are a lot of buttons that you can push, and I will leave."

**Self-harm.**    Another ineffective long-term strategy was that of self-harm. Examples of this included cutting and beating on one's chest in an attempt to self soothe. Although this may be self-soothing in the moment, they are included as ineffective strategies because they were regarded by the interviewees as unhealthy coping strategies. One example of this from Christy, a 27-year-old Asian participant, was:

> "So years ago I used to cut. And I know that that was an unhealthy coping mechanism, and it scared a lot of people because people equate cutting with suicidal ideation or suicidal behavior, paired with suicidal behavior. And so, I've worked hard to control my cutting urges. I still get urges a couple times a year, but that's different than a couple of times a day, which I had gotten many years ago."

## Discussion

Sexual trauma is incredibly common [13] and the results of this study provide insight into how individual women navigate relationships after the experience of sexual trauma. There are well-documented negative effects of sexual trauma for women [3–6]. However, far less research examines what happens when these women move into healthy sexual relationships after the experience of trauma. By interviewing women who had experienced sexual trauma and were in healthy sexual relationships at the time of the interview, we aimed to capture the nuance of strategies used to move into healthy relationships after the experience of sexual trauma. These

findings have important implications for professionals in the field, survivors' support networks, other women coping and healing from sexual trauma, and the findings provide some interesting areas for future research. The current study highlights ingenuity, self-love, and resilience in these women as they developed their strategies for coping with sexual trauma; something that is lacking in the sexual trauma literature that so often focuses on women purely as victims. Prior to this study, information regarding individual coping strategies was limited. Even more limited prior to this study were the tangible ways in which women navigate relationship building and sustainability over the longer term after experiencing sexual trauma. We do not aim to minimize the disruptive and damaging impact that sexual trauma has on women's lives. It remains a crucially important goal to minimize the perpetration of sexual violence in society. However, until that is addressed, we must not ignore the future sexual and relational lives of women who are affected by sexual trauma. Like every other human, it is their human right to experience sexual pleasure and engage in healthy relationships [38]; this paper provides 41 women's accounts of how they've navigated moving from experiencing sexual trauma toward a healthy sexual relationship.

Of the five effective strategies that were highlighted by the participants in this study, social support was by far the most important. Participants discussed how storytelling was a powerful tool in their social support networks, as it allowed them to share through social support. As an effective strategy the women noted that the identification of a "tribe" and finding others to help build that sense of community through similar experiences was critical to healing. Additionally, this social support and the ability to feel seen by others through social support led to feelings of empowerment, which were described as instrumental to moving into a healthy sexual life. Evidence from this study indicating the effectiveness of social support aligns with conclusions made in previous studies related to different types of traumatic stress and the importance of social support in coping with stressors that result from trauma [39]. Moreover, research connecting the negative effects of trauma related PTSD to negative social reactions indicates a relationship between social connections and PTSD regression, further supporting our findings related to the value of social support [15].

Communication was identified in the data as both an effective and ineffective strategy, depending on the context. For women who felt it was effective, they expressed that communicating boundaries and using that to determine who was an ally to share their experience of sexual trauma with. For women who felt communication was an ineffective strategy, the experiences centered around communicating with people who did not believe them or take them seriously. When participants felt like they were not believed or not taken seriously, it was problematic and hurtful and made communicating their experiences feel like something they shouldn't do. Research has shown that when women are not believed about their experience of sexual assault, there are negative health consequences [40, 41]. Furthermore, previous research using the same sample showed that being able to express consent and communicate one's sexual needs was important for the development of a healthy sexual relationship [42]. Other recent research has also showed that being able to express consent for sexual activity has been positively associated with sexual satisfaction [43]. However, many women have negative associations with refusing sex [44]. Thus, it is important to help women feel empowered to communicate their needs, wants, and boundaries both with sexual partners as well as with others who they may share their traumatic experiences with.

Although many women expressed how communicating to others about trauma was not always helpful, avoidance was also expressed as a barrier to moving into a healthy sexual relationship. The participants discussed several ways that avoidance was engaged in, and this overlapped with the self-harm theme. Substance use, self-harm, engaging in casual sex to numb, and avoiding communicating about feelings were all soothing for the moment but clearly

labelled as an ineffective long-term strategy for our participants. Although labelled as an ineffective strategy, participants overwhelmingly expressed a great deal of grace and understanding toward themselves for the reason they engaged in these avoidant and self-harming behaviors.

Prior research has consistently documented the effects of sexual trauma on many aspects of well-being [3–5] and the positive role that engaging in therapy has had on processing trauma. Although therapy was a successful strategy for coping with trauma for some of our participants, the results indicated that therapeutic fit was particularly important, and some women did not have positive experiences with therapy as a strategy to move beyond sexual trauma. Participants who found therapy to be ineffective mentioned discomfort with the process or specific technique and difficulty to find a suitable provider as an obstacle when attempting to utilize therapy as a coping mechanism. Participants who found therapy to be an effective strategy expressed gratitude that they had access, and this is a call to ensure greater access to therapy services, especially in the United States. Participants expressed feelings of possible judgement on the part of the therapist, and multiple women were dissatisfied with the lack of diversity as they searched for a provider who would be a good fit for their unique experience. One possible interpretation of this is that individuals are seeking professionals who can empathize with their experiences, not only as survivors of sexual assault but as women of color, queer women, and women from different socio-economic backgrounds. As many of the participants cited the benefits of therapy as an effective tool that played an important role in the coping process, taking the time to better understand the ways in which therapy is lacking for others could play a crucial role in informing effective strategies that can be accessed by diverse groups of women. This points to a real need for trained therapists who have knowledge, skills, and perhaps even lived-experiences similar to the clients they serve. The lack of diversity and inclusion and competence with working with diversity in therapists was by far the largest barrier to therapy being an effective strategy for the women in our study. Given that our study consisted of majority white participants, we would imagine this will be even more important in a more diverse sample. Indeed, a recent study found that transgender individuals were more likely to experience sexual assault compared to cisgender individuals [45]. Thus, future research should include people of diverse racial, socioeconomic background, and sexual and gender minority groups.

These interviews provided the opportunity to understand the nuanced experiences of 41 women who experienced sexual trauma and now consider themselves to be in a healthy sexual relationship. As noted above, one of the limitations of this study was that it was a primarily white sample of cisgender women in high-resource settings. It would be helpful for future research to examine experiences of resilience in trans or gender nonbinary participants and additional participants of color. We also see opportunities for future research to include the partner perspectives. Since all of our participants were in a healthy happy relationship, it would have been interesting to also collect dyadic interview data to understand the dynamics in greater depth.

In-depth qualitative research requires the consideration of research positionality; asking researchers to consider their own identities in the context of conducting the interviews and engaging in the analysis [46]. The authors involved in this research are relatively representative of the participant pool in gender (all cisgender women), race/ethnicity (three white and one Latinx researcher), and sexual identity (one bisexual, one queer, two heterosexual). Two of the authors have experienced sexual trauma themselves and two of the authors are practicing therapists who have worked with clients with prior history of sexual trauma to help them into healthy sexual relationships. We recognize these identities may introduce potential biases, emphasizing the importance of reflexivity throughout the research process to ensure the

participants' voices remained the focus of our findings rather than our own experiences [47], and we kept this at the forefront throughout the process.

To conclude, survivors of sexual trauma face many obstacles when it comes to coping and moving forward with their lives, yet the women in this study demonstrated incredible resilience while implementing strategies to cope with their trauma and went on to experience healthy fulfilling sexual lives. Our participants' experiences clearly demonstrated that sexual trauma does not equate to the end of a healthy relationship with sex or relationships. The participants detailed several strategies that were helpful and unhelpful in their process. Giving voice to survivors of sexual trauma is not only empowering for those sharing their experiences, but also for others who are earlier on in the healing process and can learn from those before them. This study is valuable as it guides researchers, health providers, therapists, survivors, and the broad community toward better understanding the processes involved in effective and ineffective strategies to moving beyond sexual trauma toward a fulfilling and healthy sexual life.

## Supporting information

**S1 Table. Representative quotes for each theme of effective and ineffective strategies.**
(DOCX)

## Author Contributions

**Conceptualization:** Kristen P. Mark.

**Data curation:** Kristen P. Mark, Laura M. Vowels.

**Formal analysis:** Kristen P. Mark, Laura M. Vowels, Lindsey Mullis, Katarina Hoskins.

**Funding acquisition:** Kristen P. Mark.

**Investigation:** Kristen P. Mark.

**Methodology:** Kristen P. Mark.

**Project administration:** Kristen P. Mark.

**Resources:** Kristen P. Mark.

**Software:** Kristen P. Mark.

**Supervision:** Kristen P. Mark.

**Validation:** Kristen P. Mark, Laura M. Vowels.

**Visualization:** Laura M. Vowels.

**Writing – original draft:** Kristen P. Mark, Laura M. Vowels, Lindsey Mullis, Katarina Hoskins.

**Writing – review & editing:** Kristen P. Mark, Laura M. Vowels, Lindsey Mullis, Katarina Hoskins.

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
