## [Decision Letter · Decision Letter 0]

29 May 2023

PONE-D-23-04051Women's strategies for navigating a healthy sex life post-sexual traumaPLOS ONE

Dear Dr. Mark,

Thank you for submitting your manuscript to PLOS ONE. After careful consideration, we feel that it has merit but does not fully meet PLOS ONE’s publication criteria as it currently stands. Therefore, we invite you to submit a revised version of the manuscript that addresses the points raised during the review process.

ACADEMIC EDITOR: 

Following Minor revisions are required:

Reviewer 01

I would like to give my feedback on your content. Firstly, it is important to include the demographics of your participants in order to provide vital information in relation to the results obtained. Additionally, it is crucial to include a discussion of the ethical considerations taken into account during your research, as well as your positionality and reflexivity in order to ensure transparency and accountability.

Furthermore, I recommend that you write a separate conclusion for your research, which should summarize your findings and provide future recommendations for further study. It is also important to address any limitations and implications of your study, as this will help to contextualize your results and provide a more comprehensive understanding of your research.

I hope these points are helpful in improving the quality of your research.

Reviewer 02

Authors did not follow the Prescribed format for Quotations in Themes like Long quotations in the Vancouver Style are held to have 40 words or more. These are laid out in a separate paragraph of text and indented clearly from the left margin. No inverted commas/quotation marks are included while Short quotations are held to be less than 40 words in the Vancouver Style required double space and inverted commas so please check this mistake and rectify it.\\ 

We look forward to receiving your revised manuscript.

Kind regards,

Sadia Jabeen, Ph.D.

Academic Editor

PLOS ONE

Additional Editor Comments:

Following Minor revisions are required:

Reviewer 01

I would like to give my feedback on your content. Firstly, it is important to include the demographics of your participants in order to provide vital information in relation to the results obtained. Additionally, it is crucial to include a discussion of the ethical considerations taken into account during your research, as well as your positionality and reflexivity in order to ensure transparency and accountability.

Furthermore, I recommend that you write a separate conclusion for your research, which should summarize your findings and provide future recommendations for further study. It is also important to address any limitations and implications of your study, as this will help to contextualize your results and provide a more comprehensive understanding of your research.

I hope these points are helpful in improving the quality of your research.

Reviewer 02

Authors did not follow the Prescribed format for Quotations in Themes like Long quotations in the Vancouver Style are held to have 40 words or more. These are laid out in a separate paragraph of text and indented clearly from the left margin. No inverted commas/quotation marks are included while Short quotations are held to be less than 40 words in the Vancouver Style required double space and inverted commas so please check this mistake and rectify it.

Reviewers' comments:

Reviewer's Responses to Questions

**Comments to the Author**

1. Is the manuscript technically sound, and do the data support the conclusions?

Reviewer #1: Partly

Reviewer #2: Yes

2. Has the statistical analysis been performed appropriately and rigorously? 

Reviewer #1: N/A

Reviewer #2: N/A

3. Have the authors made all data underlying the findings in their manuscript fully available?

Reviewer #1: No

Reviewer #2: No

4. Is the manuscript presented in an intelligible fashion and written in standard English?

Reviewer #1: Yes

Reviewer #2: No

5. Review Comments to the Author

Reviewer #1: I would like to give my feedback on your content. Firstly, it is important to include the demographics of your participants in order to provide vital information in relation to the results obtained. Additionally, it is crucial to include a discussion of the ethical considerations taken into account during your research, as well as your positionality and reflexivity in order to ensure transparency and accountability.

Furthermore, I recommend that you write a separate conclusion for your research, which should summarize your findings and provide future recommendations for further study. It is also important to address any limitations and implications of your study, as this will help to contextualize your results and provide a more comprehensive understanding of your research.

I hope these points are helpful in improving the quality of your research.

Reviewer #2: Authors did not follow the Prescribed format for Quotations in Themes like Long quotations in the Vancouver Style are held to have 40 words or more. These are laid out in a separate paragraph of text and indented clearly from the left margin. No inverted commas/quotation marks are included while Short quotations are held to be less than 40 words in the Vancouver Style required double space and inverted commas so please check this mistake and rectify it.

6. PLOS authors have the option to publish the peer review history of their article (what does this mean?). If published, this will include your full peer review and any attached files.

Reviewer #1: No

Reviewer #2: **Yes: **Dr. Ume Habiba

---

## [Author Response · Author response to Decision Letter 0]

15 Aug 2023

Response to Reviewers for PONE-D-23-04051

“Women’s strategies for navigating a healthy sex life post-sexual trauma”

Reviewer 01

I would like to give my feedback on your content. Firstly, it is important to include the demographics of your participants in order to provide vital information in relation to the results obtained. Additionally, it is crucial to include a discussion of the ethical considerations taken into account during your research, as well as your positionality and reflexivity in order to ensure transparency and accountability.

We have revised the paper to include demographic characteristics of our participants in the method section of the paper but also after each of the quotes. And thank you for bringing up positionality and reflexivity – we have added this to the discussion section as suggested and think this improves the richness of the manuscript. 

Furthermore, I recommend that you write a separate conclusion for your research, which should summarize your findings and provide future recommendations for further study. It is also important to address any limitations and implications of your study, as this will help to contextualize your results and provide a more comprehensive understanding of your research.

We have provided a separate conclusion for the research at the end of the paper and it does help to contextualize the results. Thank you for this suggestion. Additionally, we added more explicit limitations and future directions to the discussion of the paper. 

I hope these points are helpful in improving the quality of your research.

These points are very helpful in improving the quality of our research and we appreciate that you took the time to review our manuscript. 

Reviewer 02

Authors did not follow the Prescribed format for Quotations in Themes like Long quotations in the Vancouver Style are held to have 40 words or more. These are laid out in a separate paragraph of text and indented clearly from the left margin. No inverted commas/quotation marks are included while Short quotations are held to be less than 40 words in the Vancouver Style required double space and inverted commas so please check this mistake and rectify it.\\ 

Thank you for taking the time to review the paper and for this suggestion, it helps with flow of the paper to have it laid out in this way. We have made this change throughout.

---

## [Editor Report · Decision Letter 1]

21 Aug 2023

Women's strategies for navigating a healthy sex life post-sexual trauma

PONE-D-23-04051R1

Dear Dr. Mark,

We’re pleased to inform you that your manuscript has been judged scientifically suitable for publication and will be formally accepted for publication once it meets all outstanding technical requirements.

Kind regards,

Sadia Jabeen, Ph.D.

Academic Editor

PLOS ONE
---

## [Editor Report · Acceptance letter]

25 Aug 2023

PONE-D-23-04051R1 

Women’s Strategies for Navigating a Healthy Sex Life Post-Sexual Trauma 

Dear Dr. Mark:

I'm pleased to inform you that your manuscript has been deemed suitable for publication in PLOS ONE. Congratulations! Your manuscript is now with our production department. 

Kind regards, 

on behalf of

Dr. Sadia Jabeen 

Academic Editor

PLOS ONE